# CRISPR/Cas Genome Editing Technologies for Plant Improvement against Biotic and Abiotic Stresses: Advances, Limitations, and Future Perspectives

**DOI:** 10.3390/cells11233928

**Published:** 2022-12-05

**Authors:** Yaxin Wang, Naeem Zafar, Qurban Ali, Hakim Manghwar, Guanying Wang, Lu Yu, Xiao Ding, Fang Ding, Ni Hong, Guoping Wang, Shuangxia Jin

**Affiliations:** 1Hubei Key Laboratory of Plant Pathology, Huazhong Agricultural University, Wuhan 430070, China; 2Hubei Hongshan Laboratory, National Key Laboratory of Crop Genetic Improvement, Huazhong Agricultural University, Wuhan 430070, China

**Keywords:** CRISPR/Cas9, CRISPR/Cas12, CRISPR/Cas13, base editing, Prime Editing, biotic and abiotic stresses

## Abstract

Crossbreeding, mutation breeding, and traditional transgenic breeding take much time to improve desirable characters/traits. CRISPR/Cas-mediated genome editing (GE) is a game-changing tool that can create variation in desired traits, such as biotic and abiotic resistance, increase quality and yield in less time with easy applications, high efficiency, and low cost in producing the targeted edits for rapid improvement of crop plants. Plant pathogens and the severe environment cause considerable crop losses worldwide. GE approaches have emerged and opened new doors for breeding multiple-resistance crop varieties. Here, we have summarized recent advances in CRISPR/Cas-mediated GE for resistance against biotic and abiotic stresses in a crop molecular breeding program that includes the modification and improvement of genes response to biotic stresses induced by fungus, virus, and bacterial pathogens. We also discussed in depth the application of CRISPR/Cas for abiotic stresses (herbicide, drought, heat, and cold) in plants. In addition, we discussed the limitations and future challenges faced by breeders using GE tools for crop improvement and suggested directions for future improvements in GE for agricultural applications, providing novel ideas to create super cultivars with broad resistance to biotic and abiotic stress.

## 1. Introduction

Plants produce food, fuel, and feed, which are essential in daily human and animal life for nourishment and growth. In the process of plant growth, they will be affected by a variety of biological stresses (bacteria, viruses, fungi, and insects) and abiotic stresses [1,2,3,4,5]. Due to continuous global climate change and anthropogenic activity, the impact of abiotic stresses on crop growth and development is becoming more serious. Abiotic stresses, including drought, salinity, waterlogging, heat/cold, and heavy metals, significantly reduce agricultural production worldwide. Therefore, the ability to breed new species that are tolerant to various stresses in order to reduce yield loss will be a sustainable way to overcome these obstacles and meet the growing needs of human beings. Different types of biotic stresses involve a complex interplay among pathogens and host plants based on the susceptibility/resistance of crop plants to any disease. The latest advances in molecular tools have provided insights into a wide array of pathogen infection mechanisms and their interactions with specific crop plants. The insertions/deletions (Indels) mutations by artificial or natural phenomena might be involved in altering these interactions and hinder the pathways involved in the mode of infection [6,7].

Traditional crop breeding such as crossbreeding is an effective method that has been widely used to modify various plant species. Crop productivity and varieties can be increased effectively through crop breeding programs. In modern agriculture, the key methodologies used for breeding purposes are transgenic breeding, mutation-breeding, and GE-mediated breeding for crop improvement [8]. Cross-breeding and genetic recombination require years to introduce desirable alleles and increase variability [8]. Transgenic breeding is easy and well-known, improving crop traits by the exogenous transformation of genes into economically important elite varieties greatly shortens the breeding time. Still, this method inserts specific genes into the genome at random locations through plant transformation, which results in varieties containing foreign DNA. Compared to crossbreeding, mutation-breeding, and traditional transgenic breeding, GE-mediated crop breeding is fast, efficient, and accurate (Figure 1). GE improves a targeted trait in a very fast and short time and exactly revising the target gene or regulatory sequence or changing the DNA and/or RNA bases in elite varieties. The current GE technique includes meganuclease (MegN), zinc-finger nucleases (ZFNs), transcription activator-like effector nucleases (TALENs), and clustered regularly interspaced short palindromic repeats (CRISPR)/CRISPR-associated protein 9 (Cas9) (CRISPR/Cas9) [7,9,10,11]. In 2013, genetic modifications through the CRISPR/Cas9 method were developed in plants and revolutionized the field by eliminating the barriers to targeted GE. The CRISPR system has been used in wheat, rice, tobacco, and *Arabidopsis thaliana* [12,13,14,15,16,17,18,19,20,21,22,23,24,25]. Till now, GE has been practiced in more than 50 plant species, and it will revolutionize plant breeding [26,27,28,29,30,31,32,33,34,35,36,37,38,39,40] (Table 1).

Based on the composition of the CRISPR locus, this system has been divided into two classes: Class 1 requires multiple effector proteins with subtypes I, III, and IV, while class 2 requires only a single effector protein with subtypes II, V, and VI. The mode-of-action of GE by site-directed nucleases (SDNs) is that once present in a cell by insertion/expression and or transfection, the SDN is capable of cutting the genome at a targeted site. The cellular DNA-repair mechanisms fix the cut sites either by the non-homologous end joining (NHEJ) or by homology-directed repair (HDR). As NHEJ can be an error-prone process, indels can appear at the respective genomic site, leading to a loss-of-function edited gene sequence due to frameshift mutations. GE by using SDNs, can be categorized into three types: SDN-1 introduces small insertions or deletions which carry no additional or recombinant DNA. SDN-2 introduces short insertions or editing of a few base pairs by an external DNA-template sequence. The SDN-3, using a similar method to SDN-2, can be considered transgenic due to the insertion of large DNA pieces [62,63]. Since its introduction, in recent years, constant improvements have been made to make CRISPR systems easier and more suitable for different constraints, such as CRISPR/Cas9 [12,13,42,55], CRISPR/Cas12a [53,58,64,65,66], CRISPR/Cas12b [56], CRISPR/Cas13 [67,68], base editing tools [43,54,59,69,70,71,72], and CRISPR transcriptional activation (CRISPRa) [73,74,75,76,77] (Figure 2). A new form of GE technology, known as Prime Editing (PE) has recently been developed which is capable of achieving various forms of editing, for example, some base-to-base transfer, such as all transformations (C→T, G→A, A→G, and T→C) and transversion mutations (C→A, C→G, G→C, G→T, A→C, A→T, T→A, and T→G), as well as small indels without double-stranded breaks in the DNA. Since PE has enough versatility to accomplish specific forms of editing in the genome, it has great potential to grow superior crops for different purposes, including production, avoiding various biotic and abiotic stresses, and enhancing the quality of plant products [45,46,49,50,51,57,70,71,78,79,80].

CRISPR/Cas method has become the most popular among editing technologies and, thus far, has revealed the greatest potential to overcome the developing challenges (such as yield and biotic and abiotic stresses) of agriculture [9,81,82,83]. For example, mutations conferring resistance to various diseases in lettuce also exist [84]. Resistance against powdery mildew has been successfully acquired in barley by creating mutants at the mildew resistance locus o (*MLO*) [85]. The mutation at *MLO* is remarkable because it provides extraordinary, stable, and precise resistance for two decades against mildew without breakage of alleles; this long-lasting resistance is because of gene knockout [86,87]. Herein, we have summarized the recent developments and advances of CRISPR/Cas GE techniques to enhance crop resistance in biotic resistance (fungi, viruses, and bacteria) and abiotic (drought, salt, cold, and heat) resistance in sustainable agriculture, and discussed the advantages, limitations, and future prospects of the CRISPR/Cas system in modern agriculture.

## 2. CRISPR/Cas Technique for Disease Resistance

Biotic stresses, such as bacterial, viral, and fungal diseases, as well as herbivores, damage plant products every year, affecting 11% to 30% of worldwide agriculture production [88]. Plant defense against pathogens can reduce the effects of disease on plant growth and productivity, which is highly relevant to the lack of food availability in the world with the increasing population. Improvements in new methods or GE techniques have improved the new resistant crops, reducing yield losses due to plant defense. Until now, CRISPR/Cas techniques were mostly used against viral infection and for fungal and bacterial disease resistance (Figure 3). The CRISPR/Cas system has been used to develop resistance to many pathogen species [26,89] (Table 2).

### 2.1. CRISPR/Cas-Mediated Fungal Resistance in Plants

Many fungal pathogens cause lethal diseases in crop plants, such as rust, mildew, rot, and smut, which not only damage yield yearly in the biosphere but also damage the quality of the product. CRISPR/Cas has improved mycological resistance in various crop species based on the available information of the genomic mechanisms involved in crop-pathogen interactions. Defined candidate genes and gene products have provided the potential to increase plant defense against fungi [106,107]. In three crop varieties, RNA-guided Cas9 endonuclease was used to target *MLO* loci, such as tomato (*Solanum lycopersicum*), grapevine (*Vitis vinifera*), and wheat [47,100,105,110,111], and transgene-free plants have been generated [122]. An *MLO* encoded protein is localized in the cell membrane and contains seven transmembrane domains, which universally exist in all dicots and monocots [123]. Plants carrying loss-of-function alleles (*mlo*) of the *MLO*, such as *A. thaliana*, tomato, and barley, confer durable resistance against powdery mildew [124,125,126]. Using precision GE to target the *MLO-B1* locus of the wheat genome to generate a 304K deletion Tamlo-R32 mutant maintains wheat growth and yield while providing robust powdery mildew resistance [86]. Out of three *MLO* home alleles, one (*TaMLO-A1*) has been mutated by CRISPR/Cas9 in *triticum aestivum* and displayed resilient resistance against *Blumeria graminis f. sp. tritici* infection [47]. The CRISPR/Cas-mediated transgene-free and self-pollinated tomato variety, which was developed by deleting the 48 bp fragment in the *SlMlo1* gene (out of 16 important *SlMlo* genes), offers resistance against powdery mildew *Oidium neolycopersici* [105].

In grapevine, loss of *VvMLO7* function by RNAi reduced sensitivity against powdery mildew *Erysiphe necator* [127]. In parallel, the knockout of *VvMLO7* and *VvMLO3* using CRISPR/Cas9 enhanced resistance to powdery mildew in grapevine [110,111]. In apple (*Malus Domestica*) protoplasts, RNP-based technology has been successfully used to edit three (*DIPM-1*, *DIPM-2*, and *DIPM-4*) genes to create resistance against fire blight *Erwinia amylovora* [110]. CRISPR/Cas9 scheme was used to target the *VvWRKY52* transcription factor with four guide RNAs. The results showed 21% biallelic mutations in regenerated plants, and these plants confer resistance to the fungus *Botrytis cinerea* compared with monoallelic mutant plants [109]. To accelerate the GE application in woody plants, another approach based on transient leaf transformation together with disease assays was first demonstrated by researchers in *Theobroma cacao* [128]. Pathogenesis-Related 3 (*NPR3*) gene (the immune system suppressor) was targeted in cacao leaves, transiently by CRISPR/Cas9 system, so the leaves showed enhanced resistance against *Phytophthora tropicalis*. GE of a fungicide resistant gene *PcMuORP1* by CRISPR/Cas9 elucidates a novel selection marker for *Phytophthora* (a genus of oomycetes) species [129]. In rice, CRISPR/Cas9-mediated disruption of *OsSEC3A* and *OsERF922* genes confer resistance against rice blast disease [112,114]. In addition, the *pi21* gene in rice also induced durable resistance to rice blast [116]. Furthermore, resistance to *Magnaporthe oryzae* disease in rice was enhanced by generating the *OsSEC3A* mutants and showed a pleiotropic type of phenotype with an increase in salicylic acid (SA) concentration, and several genes were induced related to SA- and pathogenesis related genes [112]. To conclude, all these successful fungal disease resistance results determined the advantage, efficacy, and potential of the CRISPR/Cas-based editing system to enhance resistance in crop plants.

### 2.2. CRISPR/Cas-Mediated Viral Resistance in Plants

Plant viruses are among the most common pathogens and cause hazardous diseases in a variety of economically important crops. There are five main groups based on viral genomes characters: sense-single-stranded-RNA (ssRNA+), antisense-single stranded-RNA (asRNA-), single-stranded-DNA (ssDNA), double-stranded-DNA (dsDNA), and double-stranded-RNA (dsRNA) viruses [130]. A rolling-circle amplification system is required to replicate the virus genome through recombination-mediated duplication or by a dsDNA replicative form [131]. Their genome holds a mutual fragment of 220 bp, which is prearranged in one (A, monopartite) or two (A and B, bipartite) constituents [132]. The *Geminiviridae* are a large family (over 360 species) of ssDNA plant viruses that cause significant losses to agriculturally and economically important crop plants worldwide [131], such as *Malvaceae*, *Solanaceae*, *Fabaceae*, *Euphorbiaceae*, and *Cucurbitaceae* [133]. The commercial term for a large genus of geminiviruses is Begomoviruses. Begomoviruses mostly produce diseases in dicotyledonous plants, for example, *Nicotiana tabacum* and sweet potato (*Ipomoea batatas*), and these viruses are mostly transmitted via the whitefly or leafhopper [103,134]. CRISPR/Cas9 system was used in *Nicotiana benthamiana* and *A. thaliana* to target two different geminiviruses: Bean yellow dwarf virus (BeYDV) and Beet severe curly top virus (BSCTV), respectively [91,94]. Recently, CRISPR/Cas9 techniques have also been applied to attain resistance against Begomoviruses [90]. In the (BSCTV) genome, 43 candidates were selected to target their coding and non-coding regions using CRISPR/Cas9 [94]. In inoculated leaves, virus accumulation was significantly reduced in all CRISPR/Cas9 constructs at variable levels. However, the highest resistance was observed in *A. thaliana* and *N. tabacum* to virus infection displaying the maximum expression level of sgRNAs and Cas9. Similar results have been detected by employing 11 sgRNAs in *N. benthamiana*, targeting the non-nucleotide sequence, Rep-binding sites, Rep motifs, and the hairpin of BeYDV [91], and decreased up to 87% load of the targeted viral. A tobacco rattle virus (TRV) vector was used to deliver the sgRNA molecules to the N. benthamiana, stably overexpressing the Cas9 endonuclease to target the Tomato yellow leaf curl virus (TYLCV) genome [90]. In that study, the CRISPR/Cas approach was effectively implemented to cleave and target the virus genome during duplication to confer resistance against TYLCV [90,100,135] (Table 2).

By using specific sgRNAs, several genome loci of TYLCV (non-coding and coding sequences) were targeted in their intergenic region (IR), the RCRII motif replication protein (Rep), and the viral capsid protein (CP). Targeting the IR stem-loop invariant structure showed the lowest viral accumulation and replication [90]. A similar CRISPR/Cas9 system was established to target the geminiviruses monopartite beet curly top virus (BCTV), and bipartite Merremia mosaic virus (MerMV), which possess a similar IR stem-loop sequence. CRISPR/Cas9 system-edited BCTV and MerMV viruses displayed tempered symptoms, indicating that combined resistance against various viral strains can be achieved by a single sgRNA specific for the conserved region of the pathogen.

The traditional SpCas9 system recognizes only dsDNA, so the defense against RNA-based viruses is difficult to attain. Nevertheless, the characterization and search for associated nucleases have steered to the discovery of LwaCas13a from (*Leptotrichia wadei*) and FnCas9 from (*Francisella novicida*), which have the ability to bind and cut the RNA [102]. FnCas9 was reported to demonstrate resistance against RNA viruses [93]. The sgRNAs designed to target the RNA of cucumber mosaic virus (CMV) and tobacco mosaic virus (TMV) in *N. benthamiana* and *A. thaliana* transgenic plants showed a significant reduction in TMV and CMV by 40–80% compared to wild-type (WT) plants [93]. It demonstrated that FnCas9-mediated application could be deliberated as a CRISPR interference (CRISPRi) apparatus, similar to the mitigation of gene expression by catalytically inactive proteins of SpCas9 [136]. A similar study was carried out with Cas13a for manipulating the RNA genome of turnip mosaic virus (TuMV) using RNA-guided ribonuclease [92]. The minimum spread and replication of TuMV was observed in tobacco leaves by using the most proficient virus interference, detected with CRISPR RNA excision of *GFP2* and *HC-Pro* genes.

Furthermore, the pre-CRISPR RNA was processed by Cas13 (due to its innate ability) into functional CRISPR RNA to target many viral mRNAs simultaneously. This may provide an alternative system to improve its efficiency distinctly [92,97,137]. A second strategy is to achieve viral resistance by editing the specific plant genes that are responsible for virus resistance traits [52,95,115]. RNA viruses need plant host factors to preserve their normal life cycle, containing the eukaryotic translation initiation factors *eIF4E*, *eIF4G*, and *eIF(iso)4E* [138]. Host susceptibility gene *eIF4E* was targeted at two different sites to create resistance against plant potyviruses by CRISPR/Cas9 [52,98,99]. A similar approach in *A. thaliana* plants induced site-specific mutations at *eIF(iso)4E* locus and conferred complete resistance to single-stranded RNA potyvirus -TuMV by 1 bp deletions and 1 bp insertions without any off-target modification [95]. Recently, resistance to rice tungro spherical virus (RTSV) was developed by the mutagenesis in *eIF4G* alleles [115]. In addition, no negative effects on the growth of mutant plants were observed in studies by Macovei et al. and Pyott et al., although additional research should be conducted to verify and test the durability and efficacy of recessive resistance edited plants [95,115].

### 2.3. CRISPR/Cas-Mediated Bacterial Resistance in Plants

Many pathogenetic bacteria cause diseases in crops, and the crops show several types of symptoms [139]. Compared to fungal and viral resistance, few studies have been reported about the utilization of CRISPR/Cas against bacterial diseases in crop plant species. The *Xanthomonas oryzae pv. Oryzae* causes host gene expression to induce susceptibility by utilizing the type III transcription-activator-like effectors (TALEs) system. The *X. oryzae pv. oryzae* effector protein PthXo2 targets the host sucrose transporter gene *OsSWEET13* and is recognized as a sensitive gene for pathogen progression. Disease susceptibility was conferred by transferring the indica rice IR24 *OsSWEET13* allele to japonica rice Kitaake, while CRISPR/Cas9-mediated mutations in the allele offered resistance against bacterial blight [113]. Recently, a mutation in the promoter of three rice genes confers broad-spectrum resistance against bacterial blight in rice [117]. CRISPR/Cas9 was used to edit the promoter of the *Xa13*, a pluripotent gene for recessive resistance to bacterial blight in rice to obtain the highly resistant rice that does not affect agronomic traits [118]. Downy mildew resistance 6 (*DMR6*) is a well-known negative regulator of plant defense. In tomato, *DMR6* ortholog *SlDMR6-1* was reported to be up-expressed during *Pseudomonas syringae pv. tomato* pathogen progression and *Phytophthora capsici* infection [140]. By targeting the *SlDMR6-1* (exon-3), the mutated plants conferred wide-spectrum resistance against *P. capsici*, *Xanthomonas gardneri*, *P. syringae*, and *X. perforans* [108,140,141]. The tomato bacterial speck disease (causal agent *Pseudomonas syringae*) causes stomatal opening using coronatine (COR) to facilitate bacterial progression. This stomatal response in *A. thaliana* relies on *AtJAZ2* (Jasmonate ZIM-domain-2), a COR co-receptor. The *JAZ2* does not have the C-terminal Jas domain (*JAZ2*Δ*jas*) that avoids stomatal opening using COR [142]. The homologous gene of *AtJAZ2* in tomato is *SlJAZ2* [104]. Resistance against the model pathogen *Pseudomonas syringae pv. tomato* DC3000 (Pto) DC3000 was developed by targeting the dominant *JAZ2* repressor- *SlJAZ2*Δ*jas* by using CRISPR/Cas9 technology that prohibited stomatal opening. Improving and refining the CRISPR/Cas9 and CRISPR/Cas12a systems provide a new opportunity to edit perennial crops species such as citrus to introduce resistance against citrus greening disease [143].

After producing successful bacterial disease-resistant tomato and *A. thaliana*, the CRISPR/Cas9 system recently effectively produced citrus bacterial canker (CBC) (causal agent *Xanthomonas citri subsp. citri* (*X. citri*) resistant citrus plants. The *X. citri* is the most widespread disease in commercially cultivated citrus [41]. CBC resistance was firstly reported in Duncan grapefruit by altering the PthA4 effector binding elements in the promoter of the Lateral Organ Boundaries 1 (*CsLOB1*) gene [119]. A significant decline in Xcc symptoms was detected in the mutated lines with no additional phenotypic alterations confirming the link between CBC disease susceptibility and *CsLOB1* promoter activity Citrus (*Citrus sinensis*) (*Osbeck*) Wanjincheng orange [120]. In Wanjincheng orange, editing of *CsWRKY22* by CRISPR/Cas9 reduces susceptibility to *X. citri* [121]. CBC disease resistance was enhanced by deleting the EBEPthA4 sequence completely from both *CsLOB1* alleles, and no additional changes were observed in plants with altered *CsLOB1* promoter. Recently, the CRISPR/Cas9-FLP/FRT system has been successfully induced in apple cultivars to reduce fire blight susceptibility [144]. In conclusion, these fruitful results demonstrate that CRISPR/Cas has the potential to not only create bacterial resistance in annual and biennial crop species but also confer durable bacterial disease resistance in perennial crop plants.

## 3. CRISPR/Cas-Mediated Abiotic Stress Resistance in Plants

Abiotic stresses, such as salinity, drought, heavy metals, temperature, etc., pose a significant challenge to crop production and result in a substantial decrease in yield worldwide [145]. Climate change threatens agriculture and food security. Excessive greenhouse gas emissions are responsible for the frequent occurrence of high temperatures and drought stress in crop plants [146,147]. It is predicted that a 1 °C increase in atmospheric temperature will reduce the yield of maize, rice, and wheat by 21–31%, 10–20%, and 6%, respectively [147,148,149]. Notably, the negative effects of such abiotic stresses are more severe in South Asia and Africa, where food scarcity is already prevalent [146]. Thus, the breeding of climate-smart crops that can tolerate abiotic stresses would be a sustainable strategy for addressing these challenges.

### 3.1. CRISPR/Cas-Mediated Tolerance against Abiotic Stress in Plants

GE techniques, such as CRISPR/Cas systems, have significantly revolutionized crop improvement by enhancing resistance against abiotic stresses [145,150,151]. By activating or suppressing the target genes, GE technology is also an important tool for understanding the functions of genes involved in the resistance against abiotic stresses in plants [152,153]. CRISPR/Cas9 GE techniques have been applied to knockout the negative regulator of salt stress responses in the *A. thaliana*, *Solanum lycopersicum*, *Triticum aestivum* and *Hordeum vulgare* which are related to drought and salt stress tolerance [154,155,156,157,158,159,160,161]. Modified tomato variety lines showed highly severe symptoms on leaves (leaf wilting) in drought stress conditions compared to WT tomato plants. Knockout of Auxin Response Factor4 (SlARF4) in tomato using CRISPR/Cas9 exhibits strong salt tolerance [155]. Using CRISPR/Cas9 to generate OsDST varieties in indica mega rice cultivar MTU1010 is significant for improving drought and salt tolerance [162]. In tomato, another CRISPR/Cas9-mediated GE for heat tolerance has been accomplished by targeting the SlAGAMOUS-LIKE6 (SIAGL6) gene, resulting in enhanced fruit setting under heat stress [163]. Moreover, OsANN3 and OsMYB30 genes induce knockdown through CRISPR/Cas9 in japonica rice, which enhances the resistance mutant line against cold stress [164,165]. In a refined study, nuclease-deficient Cas9 (dCas9) or nickase Cas9 (nCas9) was fused to Petromyzon marinus cytidine deaminase (PmCDA1) to make point mutations in rice, showing resistance against herbicide in the edited plant lines [65]. In addition, mutagenesis in SiNPR1 by CRISPR/Cas9 was shown to minimize drought stress tolerance in tomato cultivars [166].

Reactive oxygen species (ROS) serve as signaling molecules to regulate gene expression and plant defense against viral pathogens and symbiotic nitrogen fixation between soil rhizobia and the plants [167,168,169]. However, overproduction of ROS, which is a typical response of plants to oxidative and abiotic stresses, can cause a variety of growth abnormalities, including a decrease in photosynthesis rate, increased cell death, and even male sterility, resulting in reduced crop yield [145]. Dozens of genes encoding antioxidant enzymes, such as glutathione S-transferases (GSTs), catalases (CATs), glutathione reductases (GRs), superoxide dismutase (SOD), and numerous peroxidases (PODs), are involved in the elimination of ROS molecules. These genes are known as the R genes and contribute to abiotic stress tolerance [170]. Molecular breeders and geneticists have identified a number of T genes related to abiotic stress tolerance and incorporated them into plants to achieve tolerance. The CRISPR/Cas9 system was recently used to develop genetic plants that constitutively overexpress the maize ARGOS8 gene by altering the natural promoter sides of the ARGOS8 gene with the GOS2 promoter [171] (Table 3). The ARGOS8 edited line showed vital alterations and improvement in grain production under field conditions using drought stress without any production drawback under natural conditions [171]. Knockout of the soybean flowering major gene GmPRR37 using CRISPR/Cas9 exhibited early flowering under natural long-day conditions, providing regionally adapted cultivars for specific regions [172]. Yield potential can be increased through manipulating an ARE1 ortholog related to nitrogen utilization efficiency in wheat by CRISPR/Cas9 [173]. Simultaneous knockout of BnaMAX1 alleles resulted in increased semi-dwarfing and branching phenotypes and more silique production, resulting in improved yield per plant relative to WT, which provides desirable germplasm for further breeding of high yield in rapeseed [174]. Moreover, contaminations of arable soils increased the heavy metals toxicity in crops. However, breeders have improved rice cultivars with a low level of arsenic, cadmium, and radioactive cesium by knocking out the *OsARM1*, *OsNRAMP5*, *OsNRAMP1*, and *OsHAK1* genes [175,176,177,178,179]. Recently CRISPR/Cas9 knockout abscisic acid receptor gene (*OsPYL*) in rice showed increased grain yield in high-temperature stress tolerance and reduced pre-harvest developing plants compared with WT [180]. Additionally, targeted mutagenesis of the OsRR22 and *OsmiR535* via CRISPR/Cas9 confers salinity tolerance, and the OsMPT3 gene is an important gene for osmotic regulation in rice [181,182,183].

### 3.2. CRISPR/Cas-Mediated Herbicide Resistance in Plants

Unwanted weeds grow everywhere in main field crops and compete with the uptake of nutrients and fertilizers. In this situation, the yield of main crops is significantly reduced, which imposes a huge loss to growers around the world. Key herbicides, such as chlorsulfuron, glufosinate, and glyphosate, as well as many selective herbicides, are involved in inhibiting the acetolactate synthase (*ALS*), 4-hydroxyphenylpyaunate dioxygenase (*HPPD*), acetyl coenzyme A carboxylase (*ACCase*), protoporphyrinogen oxidase (*PPO*), and phytoene desaturase (*PDS*) [192,193,194]. The use of excessive herbicides damages the crops due to low stress resistance against herbicide chemicals.

Currently, herbicides such as chlorsulfuron are widely used to target *ALS1* and *ALS2* genes [186]. The CRISPR/Cas9-edited *ALS1* gene increases chlorsulfuron herbicides resistance in soybean crops [61]. The rice gene *OsEPSPS* (5-enolpyruvylshikimate-3-phosphate synthase) was replaced/knocked-in using CRISPR/Cas9 to confer glyphosate resistance in plants [185]. Similarly, point mutations were introduced via CRISPR/Cas9 into the rice *ALS* gene, conferring herbicide tolerance [48]. Moreover, CRISPR/Cas9 was used to induce the loss-of-function mutation in maize *ALS2*, conferring tolerance against the herbicide [184]. A CRISPR/Cas9-mediated C287T point mutation in *ALS* resulted in an amino acid substitution of A96V and two-point mutations (G590 and W483) in *FTIP1e*, conferring resistance against imazamox [65]. Further manipulation of the CRISPR/Cas system has led to the engineering of herbicide resistance, a single mutation in the *ALS* gene as the target for base-editing [191]. Sufficient herbicide resistance is conferred in rice by designing large-scale genomic inversion or duplication using CRISPR/Cas9 [187]. The base editors ABE, CBE, and PE have recently been used to improve herbicide resistance [44,80,188,190]. Novel transgene-free herbicide-resistant watermelon varieties were created by base-editing with the potential of immediate field application to facilitate broadleaved weed control [191]. Herbicide-resistant mutants were obtained by direct evolution of the rice *OsACC* gene through dual cytosine and adenine base editors STEME-1 and STEME-NG [188]. Moreover, the M268T mutation generated in the endogenous *OsTubA2* gene by ABE endowed dinitroaniline herbicide resistance in rice without inducing fitness cost [44]. CBE was employed to target *ZmALS1* and *ZmALS2* generating sulfonylurea herbicide-resistant mutants in maize [80]. The A3A-PBE system was developed for conferring herbicide resistance in allotetraploid *Brassica napus* [190]. The above studies recommend that each crop needs a particular GE technique and perspective of genome-engineering to improve desired traits, agronomic traits, and yield under abiotic stress (Table 3).

## 4. CRISPR/Cas Systems of Advances, Limitations, and Prospective Applications

Although the CRISPR/Cas systems exhibit powerful ability in crop genetic improvement, some limitations still need to be overcome in this field [195,196,197,198,199,200]. SpCas9 requires a 5′-NGG-3ʹPAM immediately adjacent to the 20 nt DNA target sequence because it can only recognize NGG PAM sites, which limits its effectiveness. Although this restriction is vital, the goal is to turn off the gene through selective mutagenesis in any situation (Figure 4). Therefore, the main hard work to develop Cas9-like systems is underway, changing PAM sequences or causing the single CRISPR/Cas9 from *Streptococcus pyogenes* to identify other PAMs. For example, xCas9 is changed from SpCas9, which has been altered to identify a wide range of PAM sequences with GAT, NG, and GAA in mammalian cells [201]. Expanding the scope of CRISPR/Cas9-mediated GE in plants using an xCas9 and Cas9-NG hybrid [202]. A recently developed variant of SpCas9 can target an expanded set of NGN PAMs, and this enzyme was optimized for developing a near-PAMless SpCas9 variant named SpRY (NRN > NYN PAMs). SpRY nuclease and base-editor variants are capable of targeting almost all PAMs [203].

The delivery methods of CRISPR/Cas are divided into direct and indirect approaches. Direct methods include polyethylene glycol (PEG)-mediated delivery and bombardment-mediated delivery. Indirect methods include the floral dip method and *Agrobacterium tumefaciens*-mediated delivery. Direct gene delivery methods are mostly used for the transient expression of the genes. Indirect delivery based on *Agrobacterium tumefaciens*-mediated genetic transformation is mostly used in plants [204]. Nearly all GE tools in plants are based on tissue culture and the plant regeneration process. However, the regeneration of many plant species through tissue culture is a genotype dependent, time-consuming, cost-intensive, and laborious process. CRISPR/Cas GE is difficult and challenging in forest woody plants because of their lengthy propagation times, limited mutant resources, and low genetic transformation efficiency [205]. Therefore, to achieve efficient and rapid delivery of the CRISPR/Cas system to plants, the use of a suitable carrier can be considered depending on the purpose of delivery. Delivery vectors are available as plasmid and viral and non-viral vectors. Viral vectors that have been used in plants include bean yellow dwarf virus, tobacco mosaic virus, potato virus X, and cowpea mosaic virus [60,206]. However, the capacity of viral vectors limits the application of large fragment sequences or even large Cas proteins, and the use of viral vectors may stimulate the defense of the plant immune system. These non-viral vectors include a variety of materials, such as inorganic nanoparticles, carbon nanotubes, liposomes, protein- and peptide-based nanoparticles, and nanoscale polymeric materials. These novel non-viral vectors are very promising for future GE applications due to their small size, low toxicity, ability to maintain biological activity, and ability to cross many physical barriers in the domain. For citrus and grapes, Ribonucleoprotein RNP and nano-biotechnology transgene-free editing methods, and transient expression of CRISPR genes, can generate transgene-free and target gene edited plants. However, the efficiency is still low, and intensive labor is required in order to improve the current technology and develop new technologies [207].

CRISPR/Cas9 techniques can apply to other members of the kingdom Plantae, such as bryophytes, algae, and pteridophytes. The model species liverwort has emerged as an example of plant development, and the application of CRISPR/Cas9-mediated targeted mutagenesis studies has been used in the molecular breeding of liverwort Foliage (*Marchantia polymorpha* L.) [208]. Moreover, new fungus, bacteria, and virus species may be found in nature, or known ones may be sensibly changed [209].

CRISPR/Cas9 may introduce off-target mutations in plants [9,140]. Off-targets can lead to chromosomal rearrangements, causing damage at incompletely matched genomic loci, and limiting the use of GE for therapeutic purposes. Off-target effects may also lead to loss of functional gene activity, resulting in diverse physiological or signaling abnormalities [9]. Recently, whole-genome sequencing has been applied to recognize the cleavage at off-target sites by Cas9 or Cas12a system nucleases in *A. thaliana* [210], cotton [211] and rice [212]. Bioinformatics tools, such as CCTop (https://crispr.cos.uniheidelberg.de), Cas-OFFinder (http://www.rgenome.net/cas-offinder/), DISCOVER-Seq [213], Systemic evolution of ligands by exponential amplification (SELEX), Integrase-deficient lentivirus (IDLV) capture, High-throughput genomic translocation sequencing (HTGTS), and so on, have been established to manage with this issue [214]. In addition, significant advancements have been made to reduce off-target action of CRISPR/Cas9. For instance, HF-Cas9 [215], HypaCas9 [216], eSpCas9 [217], and Sniper Cas9 showed a significant reduction in off-target levels while absorbent on target action [218]. Improving current delivery methods and developing new methods will reduce barriers to the low-cost application of gene editing in crops (Figure 4). To increase the range of delivery methods, the Agrobacterium, vector and plant genes might be engineered to increase the efficacy of *Agrobacterium tumefaciens*-mediated transformation [219].

CRISPR/Cas9 system has minimal effects on the control of RNA and DNA viruses. Consequently, the advancement of an acceptable and effective CRISPR system is required to overcome such types of issues against viruses. Findings indicate that the Cas13 proteins (Cas13a, Cas13b, and Cas13c) have high potential as robust RNA regulators for RNA viruses [93]. For example, CRISPR/Cas13a conferred RNA virus resistance in monocot and dicot plants [182]. Targeted site gene editing was performed for designing eIF4E resistance alleles that play essential roles in resistance against virus [96,220], and altering the genes, which are responsible for increased metabolites (phytochemicals) that will boost abiotic and biotic stress tolerance in plants, such as drought stress tolerance, disease resistance (fungi, virus, bacteria, and phytoplasma), enhanced nutritional status, and reduced generation [221,222].

Homologous recombination (knock-in/gene replacement) mediated by CRISPR/Cas has been achieved in plants, but the editing efficiency is low [223,224]. Therefore, there is still a long way to go to achieve efficient gene knock-in by CRISPR/Cas-mediated homologous recombination in plants. The identification of more susceptible genes (S genes) in a crop genome with the new genomics strategy as the target of CRISPR systems can be achieved and remove unwanted traits [101]. On the contrary, more resistant genes (R genes) need to be cloned and knocked-in the crop genome by an improved CRISPR/Cas system via a homologous recombination-mediated DNA repair system. The molecular weight of the Cas9/Cas12a proteins is relatively large, so they cannot be packed into viral vectors for the direct delivery of Cas proteins into the plant cells without plant tissue culture. Scientists need to design several sgRNAs in one vector for multiple gene editing since the stress tolerance trait of the plant is determined by multiple genes. Accessibility of next-generation sequencing technologies will offer more adequate and accurate genome data for the assessment of target genes selection and sgRNA design in various cell types and plant species. The efficiency and accuracy of newly generated genome editing tools, including Base editors and Prime editor, are still far from satisfactory.

## 5. Conclusions and Future Perspective

Since the 1990s, various genetically modified organisms (GMO), including carrot, canola, Bt-cotton, Bt-potato, glyphosate-resistant soybean, and strawberry, have been approved to be released for beneficial usages, such as food, feed, and processing in many countries. With time, commercial cultivation of genetically modified (GM) food crops, for example, soybeans, corn, and cotton, has expanded in some countries, particularly the United States, Brazil, Argentina, India, Canada, and China. However, public skepticism about accepting GM crops is due to concerns that GM may have adverse effects on the environment or human health [225]. In China, a country with conservative attitudes towards crops and food, nearly 80% of the Chinese public accepts foods labeled as GM-free, about 40% accept GM-labeled foods, and those who are more aware of GM products are more likely to accept GM-labeled foods [226]. Using GE promises to produce crops with high yields, high quality, and good disease resistance. However, public attitudes toward GMOs suggest that people are initially unlikely to accept these plants [225]. People do not accept GE plants because they cannot tell the difference between GMO and GE plants [62,199]. GE plants alter plant traits by introducing small mutations such as deletions, insertions, and targeted mutations using CRISPR/Cas. These GE plants have resulted in significant improvements in their agronomic traits. The mutations produced by GE plants do not leave any foreign DNA behind. Additionally, the gRNA (guide RNA) used in the CRISPR/Cas system is not rDNA (recombinant DNA), so GM and GE are fundamentally different. In recent years, the use of CRISPR/Cas to generate transgene-free plants that obtain the expected agronomic traits without introducing any foreign DNA has been widely reported, thus exempting them from the definition and regulation of GMOs. Characterized by high target programmability, specificity, and robustness, CRISPR/Cas enables precise genetic manipulation of crop species, providing opportunities for creating germplasm with beneficial traits and developing novel, more sustainable agricultural systems [227]. In recent years, CRISPR/Cas has worked as a revolutionary tool with high efficiency to perform targeted GE, and it continues to progress rapidly through the invention of new CRISPR-based editing tools to achieve different goals of genome engineering, such as higher yield, pathogen-resistance, improved nutrients efficiency, and abiotic tolerance in crop species [189,228,229,230,231,232]. Many countries such as the United States, Canada, Brazil, Argentina, and Australia have exempted GE plants of SDN1-type and derived food and feed from their GMO legislation or allowed commercialization based on a simplified case-by-case procedure [233,234]. This will trigger the development of new plant varieties and a range of genome-edited plant products with minor genetic changes are expected to enter the global commodity market soon [63]. As science and technology advance, researchers will further develop various GE tools to meet people’s needs and produce high-quality and safe plants. The government must develop appropriate regulations to regulate the safety of GE plants. The government should also facilitate communication between the public and developers. If people understand the benefits of genome editing-mediated plant breeding and trust the regulations, such transgene-free plants can be gradually integrated into society. A sustainable future for agriculture can be imagined using this new and powerful GE tool.

As researchers, we must not avoid the challenges of providing clarity about CRISPR breeding methods, which promise to be significant for achieving public trust and developing regulatory policies to govern the use of the CRISPR system in agriculture. Whatever challenges remain, the newly developed CRISPR methods are just the tip of the iceberg. These powerful new plant breeding tools can provide a sustainable future for agriculture, and with that possibility comes a responsibility to alleviate the public and scientific worries regarding its usage.

## Figures and Tables

**Figure 1 cells-11-03928-f001:**
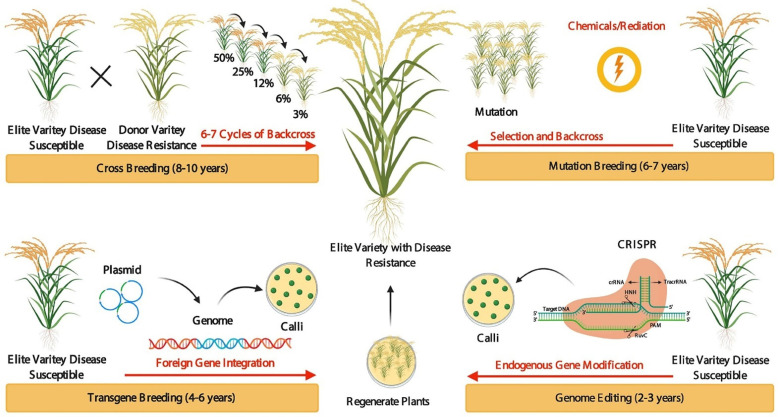
Evolution of crop breeding techniques. Crossbreeding takes a great deal of time (8–10 years) to improve desirable characters/traits (in a particular species for disease tolerance or resistance) through crossing an elite variety line with a donor variety line and selecting the new outstanding offspring with desirable characters/traits. To introduce new progeny with desirable traits from the donor variety line to the elite variety line, the selected offspring must be backcrossed to the elite variety line for several years to remove undesirable related traits. In mutation breeding, mutations are used to improve traits in the time (6–7 years) of the genome through chemical treatment or physical irradiation to create novel genetic variations. Transgenic breeding is easy and well-known, improving crop traits within (4–6 years) by the exogenous transformation of genes into economically important elite varieties. Genome editing: improving a targeted trait in a very fast and short time (2–3 years) and exactly revising the target gene or regulatory sequence or changing the DNA and/or RNA bases in elite varieties.

**Figure 2 cells-11-03928-f002:**
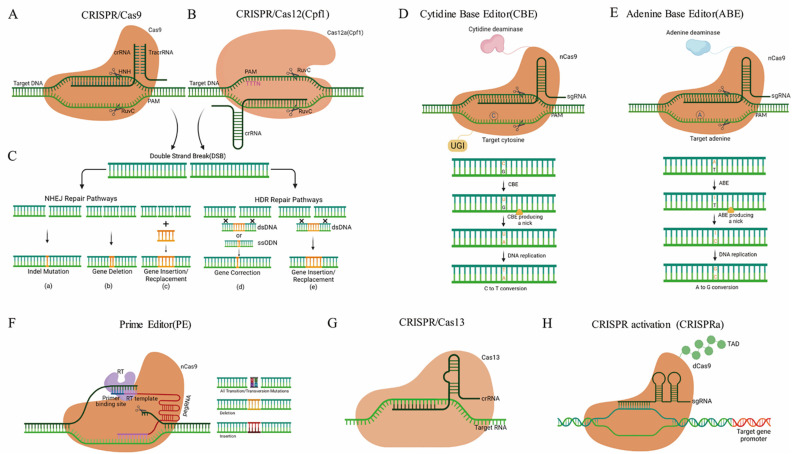
The methodology of major CRISPR/Cas systems. (**A**) CRISPR/Cas9 induces double-stranded breaks (DSBs) in DNA strands. (**B**) CRISPR/Cas12a cleaves the target DNA and introduces DSBs. (**C**) CRISPR/Cas methods can achieve different research goals: (a–c) are results of non-homologous end-joining NHEJ, and (d,e) are results of the homology-directed repair HDR repair pathways using a donor DNA template. (**D**–**F**) Base editing tools mainly include Cytidine Base Editor (CBE), Adenine Base Editor (ABE), and Prime Editor (PE). (**D**) CBE converts C-G base pairs to T-A base pairs at the target site. (**E**) ABE converts A-T base pairs to G-C base pairs at the target site. (**F**) PE is a new base editing system, which enables precise sequence substitution, insertion, and deletion. PE mainly consists of a Cas9 nickase (nCas9), an engineered reverse transcriptase (RT), and pegRNA. PegRNA includes PBS (Primer Binding Site) sequence and RT Template. (**G**) CRISPR/Cas13 consists of a Cas13, a crRNA, and a target RNA. Cas13:crRNA complexes bind target RNA and cleave the target RNA. (**H**) CRISPR transcriptional activation (CRISPRa) consists of a nuclease-deficient Cas9 (dCas9) and transcription activation domain (TAD). CRISPRa activates the transcription of single or multiple target genes.

**Figure 3 cells-11-03928-f003:**
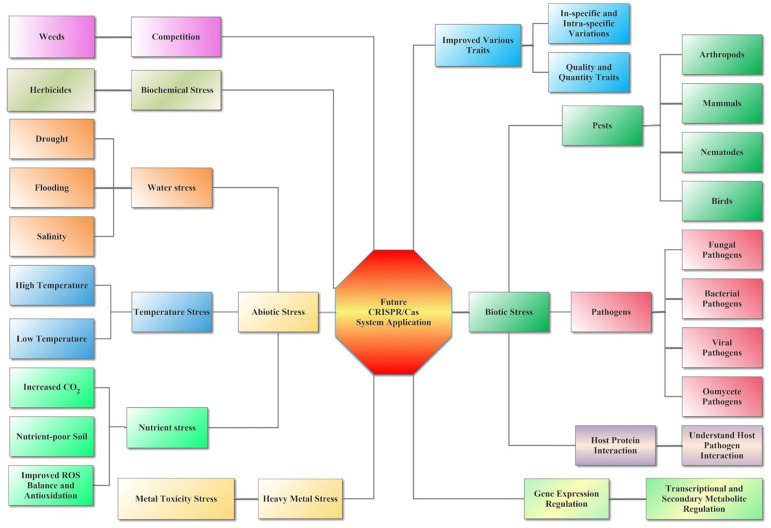
Future applications of CRISPR/Cas in plants against the biotic and abiotic stress. CRISPR/Cas represents the future of genome editing technology and the potential use of the CRISPR/Cas system in various disciplines under biotic and abiotic stresses of agriculture. With the maturity of genome editing (GE) technology and the development of new GE tools, the application of CRISPR/Cas is becoming more and more extensive. CRISPR/Cas can now achieve gene knockout, knock-in, and knock-up in plants, replacing a single base to cause amino acid changes, etc. Therefore, CRISPR/Cas can be used to modify key genes of biotic and abiotic stresses, improving crop growth and development and coping with various environmental stresses to create more germplasm resources that meet human needs.

**Figure 4 cells-11-03928-f004:**
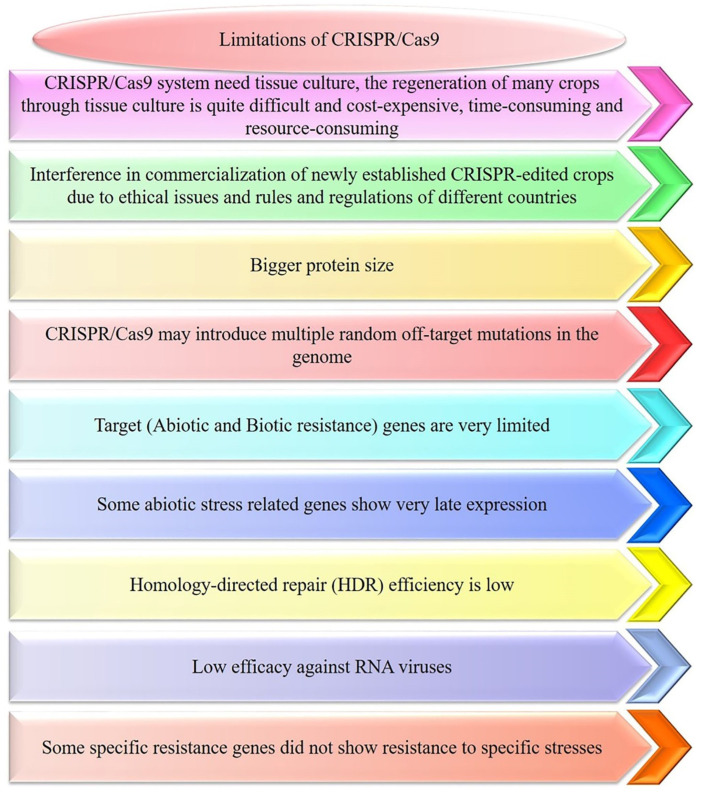
Limitations of the current CRISPR/Cas system. Using the CRISPR/Cas system in plants requires *Agrobacterium tumefaciens*-mediated transformation, but it is a time-consuming, cost-intensive, and laborious process. The selection of target genes is very limited. On the one hand, the function of resistance genes is redundant, and knocking down a gene alone cannot achieve resistance. Conversely, the knockout of resistance genes is restricted by PAM, and sequences close to PAM must be selected. CRISPR/Cas may introduce random off-target mutations in the plant genome. The commercialization of CRISPR-edited crops has been disrupted as Cas proteins take many generations to be completely isolated and obtain transgene-free crops. Currently, the homologous recombination pathway (knock-in/gene replacement) is less efficient, and the efficiency of homologous donor sequence transformation into plant cells is low, resulting in low difficulty and efficiency of knock-in. Therefore, the use of CRISPR/Cas-mediated homologous recombination in plants still has a long way to go for efficient gene knock-in.

**Table 1 cells-11-03928-t001:** CRISPR/Cas system applied in major plant species.

Plants Species	Codon-Optimization	Target Gene	Cas Promoter	sgRNA Promoter	Mutation Frequency (%)	References
*Arabidopsis thaliana*	*Arabidopsis* codon optimized	*ADH1*, *TT4*, and *RTEL1*	PcUbi4-2	AtU6-26	2.5–70.0	[28]
*Arabidopsis thaliana*	*Arabidopsis* codon-optimized	*ADH1*	PcUbi4-2	AtU6	HDR 42.8	[29,30]
*Arabidopsis thaliana*	Maize codon-optimized	*TRY*, *CPC*, and *ETC2*	2_35S	U6-26 and U6-29	42–90	[14]
*Arabidopsis thaliana*	Human codon-optimized	*FT* and *SPL4*	AtICU2	AtU6	10.00–84.78	[15]
*Arabidopsis thaliana*	*Streptococcus thermophilus* and *Staphylococcus aurous*	*ADH1*	PcUbi4-2	AtU6-26	6.1–98.5	[16]
*Citrus sinensis*	Human codon-optimized	*CsPDS*	CaMV 35S	CaMV 35 S	3.2–3.9	[41]
*Nicotiana benthamiana*	*Chlamydomonas reinhardtii* codon-optimized	*GFP*	CaMV 35S	AtU6-26	N/A	[17]
*Nicotiana benthamiana*	Plant codon-optimized	*NbFLS2* and *NbBAK1*	35S	AtU3 and AtU6	N/A	[42]
*Nicotiana benthamiana*	Plant codon-optimized	*NbPDS* and *NbIspH*	35S	AtU6-26	75–85	[18]
*Nicotiana benthamiana*	Plant and human codon-optimized	*XT*	35S	U6-26	11	[19]
*Nicotiana tabacum*	Plant codon-optimized	*NtPDS* and *NtPDR6*	2_35S	AtU6-26	81.8–87.5	[20]
*Nicotiana tabacum*	Plant codon-optimized	*mCherry*	35S-PPDK	U6	N/A	[21]
*Oryza sativa*	Rice codon-optimized	*CAO1* and *LAZY1*	OsUbi	OsU3	83–92	[22]
*Oryza sativa*	Rice codon-optimized	*OsPDS*, *OsMPK2*, and *OsBADH2*	2_35S	OsU6	HDR7.1–50	[12]
*Oryza sativa*	Plant codon-optimized	*OsBEL*	2_35S	AtU6-26	2–16	[23]
*Oryza sativa*	Rice codon-optimized	*SWEET1a*, *SWEET1b*, and *SWEET11*	OsUbi1	OsU6	12.5–100	[24]
*Oryza sativa*	Rice codon-optimized	*OsCPK6*, *OsMPK16* and *OsCPK7*	Ubi	N/A	7.69–97.92	[43]
*Oryza sativa*	Plant codon-optimized	*OsTubA2*	Ubi	OsU6	12.7	[44]
*Oryza sativa*	Plant codon-optimized	*Wx*	Ubi-1	OsU3	N/A	[45]
*Oryza sativa*	Plant codon-optimized	*OsBADH2*	Ubi	OsU6	N/A	[46]
*Sorghum bicolor*	Monocot codon-optimized synthetic	*DsRED2*	Rice Actin 1	OsU6	N/A	[17]
*Solanum lycopersicum*	Nicotiana codon-optimized	*SHR* and *SCR*	35S	AtU6	N/A	[31]
*Solanum lycopersicum*	Codon-optimized	*RIN*	Ubi4	AtU6	N/A	[32]
*Solanum lycopersicum*	Human codon-optimized	*SlPDS* and *SlPIF4*	CaMV 35S	AtUBQ and AtU6-26	72.7–100	[33]
*Triticum aestivum*	Rice codon-optimized	*TaMLO*	2_35S	TaU6	26.5–38	[34]
*Triticum aestivum*	Plant codon-optimized	*TaMLOA1*, *TaMLOB1*, and *TaMLOD1*	Ub1	TaU6	23–38	[47]
*Triticum aestivum*	Rice codon-optimized	*TaLOX2*	2_35S	TaU6	45	[34]
*Zea mays*	Plant codon-optimized	*ZmIPK*	2_35S	ZmU3	16.4–19.1	[35]
*Zea mays*	Human and maize codon-optimized	*ZmHKT1*	2_35S	Ubi1AtU6-26, OsU3, and TaU3	N/A	[14]
*Zea mays*	Maize codon-optimized	*PSY1*	ZmUbi2	ZmU6	0.18–78.83	[36]
*Zea mays*	Human codon-optimized	*Zmzb7*	2_35S	ZmU3	19–31	[37]
*Zea mays*	Maize codon-optimized	*LIG*, *MS26*, and *MS45*	Ubi	ZmU6	HDR0.13–3.9	[38,48]
*Zea mays*	Plant codon-optimized	*SHRUNKEN2*, *GBSS (WX)*	CaMV 35S	Ubi and U6-2	N/A	[49]
*Zea mays*	Plant codon-optimized	*ZmPLA1*	CaMV 35S	Ubi and U6-2	87.06	[39]
*Zea mays*	Plant codon-optimized	*ZmBADH2*	Ubi	ZmU6	N/A	[50]
*Zea mays*	Plant codon-optimized	*ZmFCP1* and *ZmCLE7*	ZmUbi	U6	N/A	[51]
*Brassica oleracea*	*Streptococcus pyogenes*	*BolC.GA4.a*	35S	U6-26	10	[40]
*Cucumis sativus*	Plant codon-optimized	*eIF4E*	35S	AtU6	N/A	[52]
*Cucumis sativus*	Plant codon-optimized	*GmPDS11* and *GmPDS18*	ZmUbi	AtU6 and GmU6	11.7–48.1	[25]
*Gossypium hirsutum*	Plant codon-optimized	*GhCLA*	Ubi	GhU6-7	1–94.12	[53]
*Gossypium hirsutum*	Plant codon-optimized	*GhCLA* and *GhPEBP*	Ubi	GhU6-7	26.67–57.78	[54]
*Gossypium hirsutum*	Plant codon-optimized	*DsRed2* and *GhCLA1*	Ubi	GhU6	66.7–100	[55]
*Gossypium hirsutum*	Plant codon-optimized	*GhCLA*	Ubi	GhU6-7	2.18–17.14	[56]
*Gossypium hirsutum*	Plant codon-optimized	*GhFAD2*	Ubi	GhU6-7	69.57	[57]
*Gossypium hirsutum*	Plant codon-optimized	*GhCLA* and *GhPGF*	Ubi	GhU6-7	68.4–89.7	[58]
*Gossypium hirsutum*	Plant codon-optimized	*GhCLA* and *GhPEBP*	CaMV 35S	GhU6-7	64	[59]
*Gossypium hirsutum*	Plant codon-optimized	*GhCLA*	CaMV 35S and Ubi	GhU6-7	44.6–97.2	[60]
*Glycine max*	Soybean codon-optimized	*DD20* and *DD43*	GmEF1A2	GmU6	HDR59–76	[61]

**Table 2 cells-11-03928-t002:** CRISPR/Cas induced plant resistance against various diseases.

Plant Species	Objective Gene	Transformation Method	CRISPR/Cas9 Induced Resistance against Plant Pathogens	References
*Nicotiana benthamiana*	*CP*, *Rep*, and *IR*	*Agrobacterium tumefaciens*-mediated transformation	Tomato Yellow Leaf Curl Virus (TYLCV) and Beet Curly Top Virus (BCTV)	[90]
*Nicotiana benthamiana*	*LIR* and *Rep/RepA*	*Agrobacterium tumefaciens*-mediated transformation	Bean Yellow Dwarf Virus (BeYDV)	[91]
*Nicotiana benthamiana*	*GFP1*, *GFP2*, *HC-Pro*, and *CP*	*Agrobacterium tumefaciens*-mediated transformation	Turnip mosaic virus (TuMV)	[92]
*Nicotiana benthamiana* and *Arabidopsis thaliana*	*ORF1*,*2*,*3*, *CP* and *30UTR*	*Agrobacterium tumefaciens*-mediated transformation	Cucumber mosaic virus (CMV) and Tobacco mosaic virus (TMV)	[93]
*Nicotiana benthamiana* and *Arabidopsis thaliana*	*CP*, *Rep*, and *IR*	*Agrobacterium tumefaciens*-mediated transformation	Bean Yellow Dwarf Virus (BeYDV)	[94]
*Arabidopsis thaliana*	*eIF(iso)4E*	*Agrobacterium tumefaciens*-mediated transformation	Turnip mosaic virus (TuMV)	[95]
*Arabidopsis thaliana*	*eIF4E1*	*Agrobacterium tumefaciens*-mediated transformation	Clover yellow vein virus (ClYVV)	[96]
*Solanum tuberosum*	*P3*, *CI*, *NIb* and *CP*	*Agrobacterium tumefaciens*-mediated transformation	Potato virus Y (PVY)	[97]
*Solanum tuberosum*	*eIF4E*	*Agrobacterium tumefaciens*-mediated transformation	Potato virus Y (PVY)	[98]
*Solanum tuberosum*	*eIF4E1*	Protoplast transformation	Potato virus Y (PVY)	[99]
*Solanum lycopersicum*	*SlPelo* and *SlMlo1*	*Agrobacterium tumefaciens*-mediated transformation	Tomato yellow leaf curl virus (TYLCV)	[100]
*Solanum lycopersicum*	*PMR4*	*Agrobacterium tumefaciens*-mediated transformation	Powdery mildew (Oidium neolycopersici)	[101]
*Ipomoea batatas*	*SPCSV-RNase3*	*Agrobacterium tumefaciens*-mediated transformation	Sweet potato chlorotic stunt virus (SPCSV) and sweet potato feathery mottle virus	[102]
*Hordeum vulgare*	*Rep*, *MP*, and *LIR*	*Agrobacterium tumefaciens*-mediated transformation	Wheat dwarf virus (WDV)	[103]
*Solanum lycopersicum*	*JAZ2*	*Agrobacterium tumefaciens*-mediated transformation	Bacterial speck disease *(Pseudomonas syringae pv.* tomato DC3000)	[104]
*Solanum lycopersicum*	*SlMlo1*	*Agrobacterium tumefaciens*-mediated transformation	Powdery mildew (*Oidium neolycopersici)*	[105]
*Solanum lycopersicum*	*PL*	*Agrobacterium tumefaciens*-mediated transformation	Fungal disease (*Botrytis cinerea*)	[106]
*Solanum lycopersicum*	*ACET1a and ACET1b*	*Agrobacterium tumefaciens*-mediated transformation	Fungal disease (*Botrytis cinerea*)	[107]
*Solanum lycopersicum*	*SlDMR6*	*Agrobacterium tumefaciens*-mediated transformation	Broad-spectrum disease resistance	[108]
*Vitis vinifera*	*WRKY52*	*Agrobacterium tumefaciens*-mediated transformation	Gray mold (*Botrytis cinerea*)	[109]
*Vitis vinifera*	*MLO-7*	PEG-mediated protoplast transformation	Powdery mildew (*Erysiphe necator*)	[110]
*Vitis vinifera*	*VvMLO3*	*Agrobacterium tumefaciens*-mediated transformation	Powdery mildew (*Erysiphe necator*)	[111]
*Oryza sativa*	*SEC3A*	Protoplast transformation with Cas9/gRNA expression binary	Rice blast disease *(Magnaporthe oryzae)*	[112]
*Oryza sativa*	*SWEET13*	*Agrobacterium tumefaciens*-mediated transformation	Bacterial blight (*Xanthomonas oryzae p v.oryzae)*	[113]
*Oryza sativa*	*OsSWEET11* and *OsSWEET14*	*Agrobacterium tumefaciens*-mediated transformation	Bacterial blight (*Xanthomonas oryzae p v.oryzae)*	[17]
*Oryza sativa*	*OSERF922*	*Agrobacterium tumefaciens*-mediated transformation	*Rice Blast Magnaporthe oryzae*	[114]
*Oryza sativa*	*eIF4G*	*Agrobacterium tumefaciens*-mediated transformation	Rice tungro spherical virus (RTSV)	[115]
*Oryza sativa*	*Bsr-d1*, *Pi21* and *ERF922*	*Agrobacterium tumefaciens*-mediated transformation	Rice blast and bacterial blight	[116]
*Oryza sativa*	*SWEET11*, *SWEET13*, and *SWEET14*	*Agrobacterium tumefaciens*-mediated transformation	Bacterial blight *Xanthomonas oryzae pv. Oryzae*	[117]
*Oryza sativa*	*Xa13* *promoter*	*Agrobacterium tumefaciens*-mediated transformation	Bacterial blight *Xanthomonas oryzae pv. Oryzae*	[118]
*Triticum aestivum*	*TaMlo1*	*Agrobacterium tumefaciens*-mediated transformation	Powdery mildew (*Blumeria graminis f. sp. Tritici)*	[47]
*Triticum aestivum*	*TaEDR1*	*Agrobacterium tumefaciens*-mediated transformation	Powdery mildew *(Blumeria graminis f. sp. Tritici)*	[79]
*Citrus sinensis*	*LOB1*	*Agrobacterium tumefaciens*-mediated transformation	Citrus canker (*Xanthomonas citri* subspecies citric)	[119]
*Citrus sinensis*	*Phytoene desaturase (CsPDS CsLOB1)* promoter	*Agrobacterium tumefaciens*-mediated transformation	*(Carotenoid biosynthesis)* Citrus canker resistance	[120]
*Citrus sinensis*	*CsWRKY22*	*Agrobacterium tumefaciens*-mediated transformation	Citrus canker *Xanthomonas citri subsp. Citri*	[121]
*Malus domestica*	*DIPM-1DIPM 2DIPM-4*	PEG-mediated protoplast transformation	Fire blight (*Erwinia amylovora*)	[110]

**Table 3 cells-11-03928-t003:** CRISPR/Cas induced resistance against abiotic stress.

Plant Species	Objective Gene	Transformation Methods	CRISPR/Cas9 Induced Resistance in Plant against Herbicide and Abiotic Stress	References
*Solanum lycopersicum*	*SlMAPK3*	*Agrobacterium tumefaciens*-mediated transformation	Drought resistance	[154]
*Solanum lycopersicum*	*SlARF4*	*Agrobacterium tumefaciens*-mediated transformation	Salinity and Osmotic tolerance	[155]
*Solanum lycopersicum*	*SlHyPRP1*	*Agrobacterium tumefaciens*-mediated transformation	salt stress-tolerant	[156]
*Solanum lycopersicum*	*SlAGAMOUS-LIKE 6*	*Agrobacterium tumefaciens*-mediated transformation	Heat resistance	[163]
*Zea mays*	*ALS2*	*Agrobacterium tumefaciens*-mediated transformation	Herbicide resistance	[184]
*Zea mays*	*ZmALS1*, *ZmALS2*	PEG-mediated protoplast transformation	Herbicide resistance	[80]
*Zea mays*	*ARGOS8*	*Agrobacterium tumefaciens*-mediated transformation	Improve yield under drought resistance	[171]
*Arabidopsis thaliana*	*OST2*	*Agrobacterium tumefaciens*-mediated transformation	Reduced transpiration, stomatal closure, and abiotic stress	[150]
*Arabidopsis thaliana*	*UGT79-B2*, and *B3*	*Agrobacterium tumefaciens*-mediated transformation	Oxidative stress, salt and cold tolerance	[159]
*Arabidopsis thaliana*	*AVP1*	*PEG*-mediated transformation	Drought tolerance	[77]
*Oryza sativa*	*OsEPSPS*	Particle bombardment transformation	glyphosate resistance	[185]
*Oryza sativa*	*ALS*	*Agrobacterium tumefaciens*-mediated transformation	Herbicide tolerant	[186]
*Oryza sativa*	*ALS-FTIP1e*	*Agrobacterium tumefaciens*-mediated transformation	Imazamox herbicide resistance	[65]
*Oryza sativa*	*OsSAPK2*	*Agrobacterium tumefaciens*-mediated transformation	Drought tolerance	[153]
*Oryza sativa*	*OsAnn3*	*Agrobacterium tumefaciens*-mediated transformation	Cold resistance	[145]
*Oryza sativa*	*OsRR22*	*Agrobacterium tumefaciens*-mediated transformation	Salinity tolerance	[182]
*Oryza sativa*	*OsDST*	*Agrobacterium tumefaciens*-mediated transformation	Drought and salt tolerance	[162]
*Oryza sativa*	*OsbHLH024*	*Agrobacterium tumefaciens*-mediated transformation	Salt stress resistance	[157]
*Oryza sativa*	*OsGTγ-2*	*Agrobacterium tumefaciens*-mediated transformation	Salt stress resistance	[158]
*Oryza sativa*	*OsmiR535*	*Agrobacterium tumefaciens*-mediated transformation	Drought and salinity stress tolerance	[183]
*Oryza sativa*	*PPO1* and *HPPD*	PEG-mediated protoplast transformation	Herbicide resistance	[187]
*Oryza sativa*	*OsACC*	*Agrobacterium tumefaciens*-mediated transformation	Herbicide resistance	[188]
*Oryza sativa*	*OsTubA2*	*Agrobacterium tumefaciens*-mediated transformation	Dinitroaniline herbicide resistance	[44]
*Oryza sativa*	*OsMYB30*	*Agrobacterium tumefaciens*-mediated transformation	Cold tolerance	[165]
*Oryza sativa*	*OsHAK1*	*Agrobacterium tumefaciens*-mediated transformation	Heavy metal pollution resistance	[175]
*Oryza sativa*	*OsNRAMP5*	*Agrobacterium tumefaciens*-mediated transformation	Heavy metal pollution resistance	[176]
*Oryza sativa*	*OsNRAMP1*	*Agrobacterium tumefaciens*-mediated transformation	Heavy metal pollution resistance	[177,178]
*Triticum aestivum*	*TaALS-P174*	particle bombardment transformation	Herbicide Resistance	[189]
*Triticum aestivum*	*TaHAG1*	*Agrobacterium tumefaciens*-mediated transformation	Salt tolerance	[160]
*Hordeum vulgare*	*ITPK*	*Agrobacterium tumefaciens*-mediated transformation	Salt stress resistance	[161]
*Solanum lycopersicum*	*SlNPR1*	*Agrobacterium tumefaciens*-mediated transformation	Drought tolerance	[166]
*Brassica napus*	Two *BnaMAX1* homologs	*Agrobacterium tumefaciens*-mediated transformation	Increases yield	[174]
*Brassica napus*	*ALS*	*Agrobacterium tumefaciens*-mediated transformation	Herbicide resistance	[190]
*Glycine max*	*GmPRR37*	*Agrobacterium tumefaciens*-mediated transformation	Regional adaptation	[172]
*Citrullus lanatus*	*ALS*	*Agrobacterium tumefaciens*-mediated transformation	Bensulfuron herbicide resistance	[191]

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
