# Peer review of "CRISPR/Cas Genome Editing Technologies for Plant Improvement against Biotic and Abiotic Stresses: Advances, Limitations, and Future Perspectives"

_cells, 2022, doi:10.3390/cells11233928_

Round 1
Reviewer 1 Report
I have gone through the manuscript by Wang et al., overall, it is an interesting and comprehensives review about the future promise and stumbling blocks of this technique, with sufficient details about its applications including interesting diagrams. The explanation of the plant improvement against biotic and abiotic stresses is adequate and understandable (for readers who already have background in genetics). I recommend for minor revision.
In the conclusion section, maybe you could include some discussion about consumer acceptance. Are anti-GMO activists generally opposed to CRISPR-Cas9 editing or do they distinguish between the idea of editing within a plant's own genome as opposed to introducing genetic material from an external source (as in Bt corn)?
Authors should also cite few recent references given below which can improve the manuscript.
https://www.frontiersin.org/articles/10.3389/fgeed.2022.987817/full
https://www.sciencedirect.com/science/article/abs/pii/S0304423821005835
https://link.springer.com/article/10.1007/s11033-022-07391-4
Author Response
Dear editor and reviewers, Nov. 28, 2022
On behalf of all co-authors, we thank you very much for giving us an opportunity to revise this manuscript and we are truly grateful to reviewers’ critical comments and thoughtful suggestions. Based on these comments and suggestions, we have made careful modifications on the original manuscript entitled ‘CRISPR/Cas genome editing technologies for plant improvement against biotic and abiotic stresses: advances, limitations, and future perspectives’ and the authors carefully responded each comment and suggestion.
All authors listed in present manuscript had been connected to discuss and revise this manuscript. The major amendments were made by using the track changes mode in this revised manuscript. We wish this new version manuscript could meet the journal’s standard.
Point to point responses to reviewers’ comments are as follow:
Reviewer 1:
I have gone through the manuscript by Wang et al., overall, it is an interesting and comprehensives review about the future promise and stumbling blocks of this technique, with sufficient details about its applications including interesting diagrams. The explanation of the plant improvement against biotic and abiotic stresses is adequate and understandable (for readers who already have background in genetics). I recommend for minor revision.
In the conclusion section, maybe you could include some discussion about consumer acceptance. Are anti-GMO activists generally opposed to CRISPR-Cas9 editing or do they distinguish between the idea of editing within a plant's own genome as opposed to introducing genetic material from an external source (as in Bt corn)?
Response: Thanks, as suggested, we added some discussions about this point in ‘Since the 1990s, various genetically modified organism (GMO)…… sustainable future for agriculture can be imagined using this new and powerful GE tool’.
Authors should also cite few recent references given below which can improve the manuscript.
https://www.frontiersin.org/articles/10.3389/fgeed.2022.987817/full
https://www.sciencedirect.com/science/article/abs/pii/S0304423821005835
https://link.springer.com/article/10.1007/s11033-022-07391-4
Response: Thanks, as suggested, we added these reference in ‘In recent years, CRISPR/Cas has worked …… improved nutrients efficiency, and abiotic tolerance in crop species [175-180]’.
Other modifications:
- Some new references were added.
- Other format changes.
Sincerely yours,
Dr. Shuangxia Jin
Full Professor,
College of Plant science and Technology,
Hubei Hongshan Laboratory, National Key Laboratory of Crop Genetic Improvement, Huazhong Agricultural University, Wuhan City, Hubei Province, China
Email: jsx@mail.hzau.edu.cn
Reviewer 2 Report
1. This review has collection of information and represented in tables form and figures. But at the same time manuscript is not written very well.
2. Several places in this manuscript have difficulties reading and understanding the language that the authors are trying to explain.
3. They should avoid phrases like "for the first time in history" (which people have been doing for few years), and "in this review" (line 122 in the manuscript).
4. In the introduction part, they focused on food security, using outdated and irrelevant references [for example, lines 44 and 145].
5. Meganuclease was not mentioned in lines 63-69.
6. I believe they should explain SDN 1 to 3
7. They must also explain and provide recent update on crisper system variants for example type I to type III
8. Whole manuscript has serious flaws in writing genes and organism names, which must be italic or according to journal information. Authors must correct those.
Author Response
Dear editor and reviewers, Nov. 28, 2022
On behalf of all co-authors, we thank you very much for giving us an opportunity to revise this manuscript and we are truly grateful to reviewers’ critical comments and thoughtful suggestions. Based on these comments and suggestions, we have made careful modifications on the original manuscript entitled ‘CRISPR/Cas genome editing technologies for plant improvement against biotic and abiotic stresses: advances, limitations, and future perspectives’ and the authors carefully responded each comment and suggestion.
All authors listed in present manuscript had been connected to discuss and revise this manuscript. The major amendments were made by using the track changes mode in this revised manuscript. We wish this new version manuscript could meet the journal’s standard.
Point to point responses to reviewers’ comments are as follow:
Reviewer 2:
- This review has collection of information and represented in tables form and figures. But at the same time manuscript is not written very well.
Response: Thanks, as suggested, we revised our manuscript.
- Several places in this manuscript have difficulties reading and understanding the language that the authors are trying to explain.
Response: Thanks, as suggested, we revised our manuscript.
- They should avoid phrases like "for the first time in history" (which people have been doing for few years), and "in this review" (line 122 in the manuscript).
Response: Thanks, as suggested, we deleted " For the first time in the history……. several annual and biennial plants", we changed "in this review" to "here/herein".
- In the introduction part, they focused on food security, using outdated and irrelevant references [for example, lines 44 and 145].
Response: Thanks, as suggested, in the introduction part, we deleted ‘The world population is predicted …… driving the global demand for food.’ ‘According to …… to global food security at present.’ In line 145, we changed this sentence to ‘Biotic stresses, such as bacterial, viral, and fungal diseases, as well as herbivores damage plant products every year, affecting 11% to 30% of worldwide agriculture production’ and modified these references.
- Meganuclease was not mentioned in lines 63-69.
Response: Thanks, as suggested, we changed it to ‘The current GE technique includes meganuclease (MegN), zinc-finger nucleases (ZFNs), transcription activator-like effector nucleases (TALENs), and clustered regularly interspaced short palindromic repeats (CRISPR)/CRISPR-associated protein 9 (Cas9) (CRISPR/Cas9)’.
- I believe they should explain SDN 1 to 3
Response: Thanks, as suggested, we added that ‘Based on the composition of the CRISPR locus, this system has been divided into two classes: Class 1 requires multiple effector proteins having subtypes I, III, and IV, while class 2 requires only a single effector protein having subtypes II, V, and VI’.
- They must also explain and provide recent update on crisper system variants for example type I to type III.
Response: Thanks, as suggested, we added this part of the content in ‘The mode-of-action of GE by site-directed nucleases (SDNs) is……large DNA pieces’
- Whole manuscript has serious flaws in writing genes and organism names, which must be italic or according to journal information. Authors must correct those.
Response: Thanks, as suggested, we carefully reviewed the writing problems and corrected them.
Other modifications:
- Some new references were added.
- Other format changes.
Sincerely yours,
Dr. Shuangxia Jin
Full Professor,
College of Plant science and Technology,
Hubei Hongshan Laboratory, National Key Laboratory of Crop Genetic Improvement, Huazhong Agricultural University, Wuhan City, Hubei Province, China
Email: jsx@mail.hzau.edu.cn
Reviewer 3 Report
Comments on cells-2030595
In this review manuscript, the authors reviewed the recent progress of CRISPR/Cas genome editors for applications in plant genome editing. This is an important topic that few papers have reviewed so far. The logic of this review paper is clear, and most recent papers were cited and reviewed. Overall, this is a good review and suitable for publication after addressing one comment: The authors should add a brief paragraph or section to discuss the current approaches and limitations of the delivery methods, including viral and non-viral ones, for genome editing in plants.
Author Response
Dear editor and reviewers, Nov. 28, 2022
On behalf of all co-authors, we thank you very much for giving us an opportunity to revise this manuscript and we are truly grateful to reviewers’ critical comments and thoughtful suggestions. Based on these comments and suggestions, we have made careful modifications on the original manuscript entitled ‘CRISPR/Cas genome editing technologies for plant improvement against biotic and abiotic stresses: advances, limitations, and future perspectives’ and the authors carefully responded each comment and suggestion.
All authors listed in present manuscript had been connected to discuss and revise this manuscript. The major amendments were made by using the track changes mode in this revised manuscript. We wish this new version manuscript could meet the journal’s standard.
Point to point responses to reviewers’ comments are as follow:
Reviewer 3:
In this review manuscript, the authors reviewed the recent progress of CRISPR/Cas genome editors for applications in plant genome editing. This is an important topic that few papers have reviewed so far. The logic of this review paper is clear, and most recent papers were cited and reviewed. Overall, this is a good review and suitable for publication after addressing one comment: The authors should add a brief paragraph or section to discuss the current approaches and limitations of the delivery methods, including viral and non-viral ones, for genome editing in plants.
Response: Thanks, as suggested, we added this section in ‘The delivery methods of CRISPR/Cas are divided ……. develop new technologies’.
Other modifications:
- Some new references were added.
- Other format changes.
Sincerely yours,
Dr. Shuangxia Jin
Full Professor,
College of Plant science and Technology,
Hubei Hongshan Laboratory, National Key Laboratory of Crop Genetic Improvement, Huazhong Agricultural University, Wuhan City, Hubei Province, China
Email: jsx@mail.hzau.edu.cn
Round 2
Reviewer 2 Report
I'm satisfied with the author's response. Please enlarge the font size for the figures' text.